# DNA origami scaffold for studying intrinsically disordered proteins of the nuclear pore complex

Philip Ketterer[1], Adithya N. Ananth[2], Diederik S. Laman Trip[2], Ankur Mishra[3], Eva Bertosin[1], Mahipal Ganji[2], Jaco van der Torre[2], Patrick Onck[3], Hendrik Dietz[1] & Cees Dekker[2]

The nuclear pore complex (NPC) is the gatekeeper for nuclear transport in eukaryotic cells. A key component of the NPC is the central shaft lined with intrinsically disordered proteins (IDPs) known as FG-Nups, which control the selective molecular traffic. Here, we present an approach to realize artificial NPC mimics that allows controlling the type and copy number of FG-Nups. We constructed 34 nm-wide 3D DNA origami rings and attached different numbers of NSP1, a model yeast FG-Nup, or NSP1-S, a hydrophilic mutant. Using (cryo) electron microscopy, we find that NSP1 forms denser cohesive networks inside the ring compared to NSP1-S. Consistent with this, the measured ionic conductance is lower for NSP1 than for NSP1-S. Molecular dynamics simulations reveal spatially varying protein densities and conductances in good agreement with the experiments. Our technique provides an experimental platform for deciphering the collective behavior of IDPs with full control of their type and position.

---

[1] Physik Department and Institute for Advanced Study, Technische Universität München, Am Coulombwall 4a, Garching bei München D-85748, Germany.
[2] Department of Bionanoscience, Kavli Institute of Nanoscience, Delft University of Technology, Van der Maasweg 9, 2629 HZ Delft, The Netherlands.
[3] Zernike Institute for Advanced Materials, University of Groningen, Nijenborgh 4, 9747AG Groningen, The Netherlands. These authors contributed equally: Philip Ketterer and Adithya N. Ananth. Correspondence and requests for materials should be addressed to H.D. (email: dietz@tum.de) or to C.D. (email: c.dekker@tudelft.nl)

Nuclear pore complexes (NPCs) mediate all transport to and from the nucleus in eukaryotic cells. A single NPC is a complex protein structure consisting of hundreds of proteins called nucleoporins (Nups), which comprise both structural Nups that build the scaffolding structure of the NPC, and intrinsically disordered Nups[1–4]. The latter so-called FG-Nups contain hydrophobic phenylalanine–glycine repeats and are located inside the central NPC channel. The FG-Nups are responsible for the remarkable selective permeability of NPCs[5]. Several models have been proposed for the transport mechanism through NPCs, but, despite much research on the structure and function of NPCs, no consensus has been reached[6–11].

Owing to the huge (60–125 MDa) size and complexity of the NPC, deciphering its structural and functional properties represents a significant challenge. Probing and manipulating NPC transport in vivo is challenging given the complex cellular environment and the demand for true nanoscale resolution. Full in vitro reconstitution of the large NPCs would be beneficial as a much larger set of analytical methods could be employed, but has so far not been found to be feasible. Interestingly, various groups have developed biomimetic NPCs where a single type of FG-Nup is attached to nanopores within a polymeric or solid-state SiN membrane[12–14]. While this approach has provided encouraging results for NPC studies, all such previous work relied on random attachment of FG-Nups on nanopore surfaces which inherently precludes full control of the exact number, density, position, and composition of the FG-Nups.

Here we present biomimetic NPCs that provide superior control over the positioning of NPC components, based on DNA origami scaffolds[15]. DNA origami structures have previously been constructed for usage as pores and channels in lipid membranes[16–18] and also as addressable adapters for solid-state nanopores[19,20]. DNA origami technology can also be employed to create ring-like objects with custom-designed curvature[21]. Such rings have previously been employed to template liposome assembly[22]. Our DNA origami-based NPC mimic features a custom-designed multilayer DNA origami structure that resembles the ring-like shape and diameter of the NPC scaffold. Onto this scaffold, we attach yeast NSP1, an archetypal well-studied FG-Nup, at a number of defined locations on the inner ring surface. With this DNA origami scaffold approach, we gain control over the precise number and the position of the FG-Nup attachment points to affect the density of the Nups in the NPC mimic, as the user can choose where exactly to attach what type of Nup. Next to wild-type NSP1, we also study a mutant Nup, NSP1-S, where the hydrophobic amino acids F, I, L, and V were replaced with hydrophilic S[23] (see Supplementary Note 1 for sequences). We report the design of these DNA origami-based NPC mimics and present electron microscopy, ionic conductance measurements, and molecular dynamics (MD) simulations that characterize their structural and transport properties. Taken together, the data establish these DNA origami scaffolds as a promising platform for studying the NPC.

## Results

**Characterization of DNA origami rings for Nups attachment.** The origami scaffold (Fig. 1; design details in Supplementary Figures 1–2 and Supplementary Tables 1–3) consists of 18 helices that form a ring with an inner diameter of ~34 nm, which approximates the inner diameter of the central channel of NPCs[4,24]. The ring can host up to 32 attachment sites pointing radially inward. We designed 2 variants of rings, 1 with 8 and 1 with 32 attachment sites, where these copy numbers were inspired by multiple-of-8 protein abundances in NPCs. The attachment anchors contain single-stranded DNA overhangs that

can hybridize to targets, which are complementary sequence oligomers that are covalently bound to a Nup. Each attachment anchor is based on two DNA single-strands protruding from the ring which can partly hybridize in order to form a short double-helical "separator" domain (5× G-C bp) away from the ring (Supplementary Figure 1e) from which the single-strand anchor emerges. The separator part biases the orientation of the Nup attachment anchors toward the radially inward direction and thereby increases the accessibility for target attachment. To facilitate electrophoretically driven docking of the ring to solid-state nanopores, we also mounted a double-stranded DNA leash at the bottom of the ring[25]. Electrophoretic mobility analysis (EMA) was used to verify the ring assembly (Supplementary Figure 3)[26].

To probe whether the attachment anchors indeed successfully hybridizes DNA oligomers, we incubated rings with 8 and 32 attachment sites with a complementary oligonucleotide labeled with cyanine-5 (Cy5) dye and analyzed the samples using EMA (Fig. 2a). The obtained fluorescence intensity in the Cy5 channel strongly increased with the number of attachment points, yielding a significantly larger (3.2-fold higher) intensity for 32 versus 8 attachment sites. For a quantitative estimate, we counted the number of bleaching steps in TIRF fluorescence microscopy recordings on rings near a surface, which report the number of attached strands in individual NPC rings (Fig. 2b and Methods section). For rings with 8 attachment sites, we obtained a skewed distribution with a peak around 7 Cy5 molecules, a tail at lower numbers, and almost no recordings of more than 8 steps. We conclude that the large majority of the targets are successfully incorporated to the attachment anchors.

**Electron microscopy reveals different Nup densities.** For the attachment of Nup proteins to the ring, we conjugated NSP1 and NSP1-S with an oligonucleotide with the respective complementary sequence (Methods section). We incubated rings with 32 attachment anchors with NSP1 and NSP1-S (hereafter denoted as '32-NSP1' or '32-NSP1-S') and purified samples from excess protein. We employed negative stain transmission electron microscopy (TEM) to obtain images of rings without protein, 32-NSP1 and 32-NSP1-S (Fig. 2c–e). Images of bare rings without

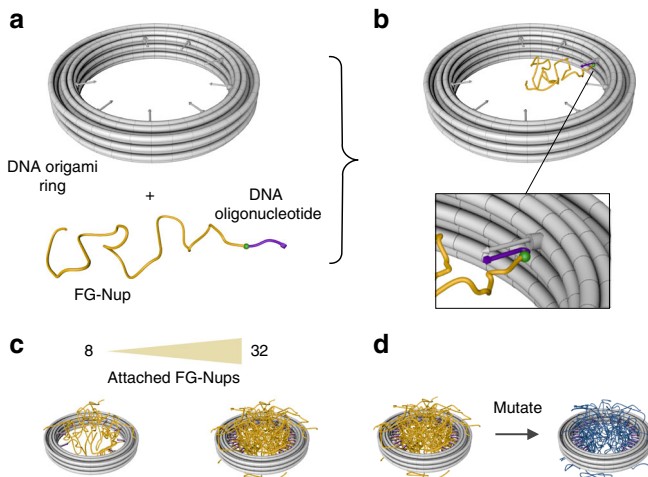

**Fig. 1** Schematics of DNA origami ring with attached FG-Nups (**a**), DNA ring (see Supplementary Figure 1 for design details) and one NSP1 protein with a covalently attached oligonucleotide. **b** Ring with attached NSP1 protein. **c** DNA ring versions with 8 (left) and 32 (right) NSP1 proteins attached. **d** DNA ring with 32-NSP1 (left) and 32 mutated NSP1-S (right) attached

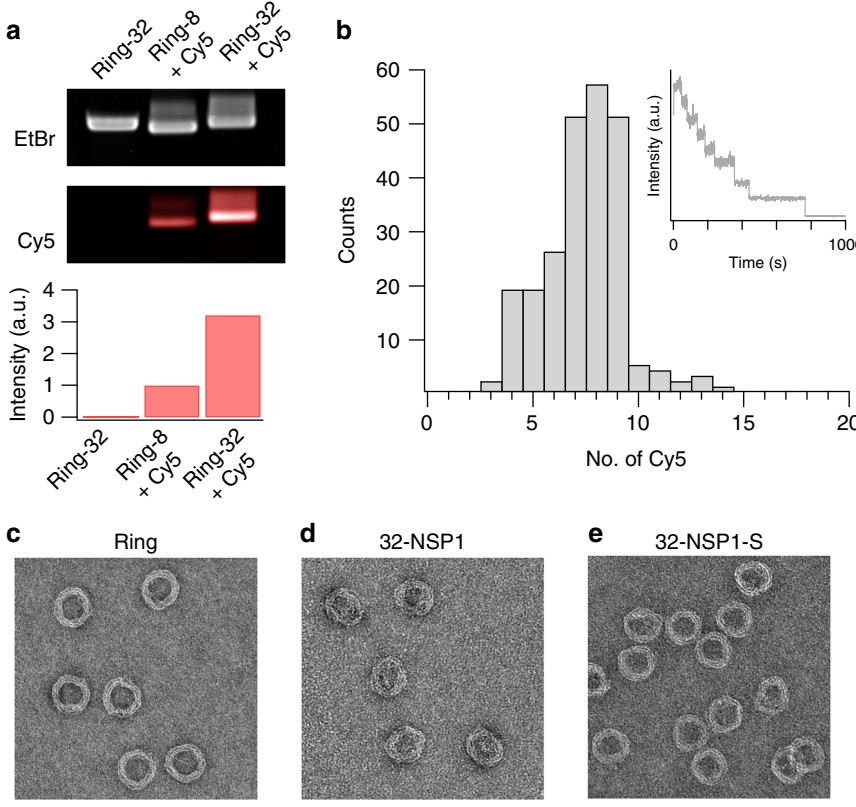

**Fig. 2** Attachment of NSP1 and NSP1-S to the DNA ring. Top: laser-scanned EtBr images of (**a**) 1% agarose gel on which DNA origami ring samples were electrophoresed. Middle: same gel scanned in Cy5 excitation/emission channels. Cy5-labeled DNA strands were added to FG-Nup attachments where indicated. Bottom: integrated Cy5 band intensity normalized to EtBr intensity. **b** Histogram of the number of attached Cy5-labeled oligomers for an 8-attachment ring. Inset shows an exemplary intensity trace of a single-particle recording of a DNA ring with 8 attachment sites incubated with the complementary Cy5-labeled oligonucleotide obtained using total internal reflection microscopy (TIRF) (Methods section and Supplementary Figure 4). See Supplementary Figure 5 for additional intensity traces. **c–e** Exemplary field-of-view negative staining TEM micrographs of DNA origami NCP-mimic ring without protein, with 32-NSP1, and 32-NSP1-S, respectively. See Supplementary Figure 6 for exemplary particles. Scale bar = 50 nm

proteins yielded well-defined particles with circular stripes corresponding to layers of DNA helices within the ring. 32-NSP1 frequently showed rings with a heterogeneous density of proteins inside and less well visible circular stripes. We attribute this to the presence of NSP1 protein that spreads out across the top of the rings (in accordance with the MD simulations discussed below). Rings incubated with NSP1-S show a lower protein density inside the rings, while also exhibiting less well visible circular stripes compared to the rings without proteins.

In addition, tomograms obtained from tilt series of negative stain electron micrographs on 32-NSP1 showed a high density in all slices along the height of the ring, indicating a rather homogenous filling of the rings with NSP1 (Supplementary Figure 7). Taken together, these results confirm the successful attachment of the proteins to the DNA origami NPC mimic. The intrinsically disordered FG-Nups appear to form a cohesive protein mass inside the origami scaffolds. As the staining agent as well as drying artifacts might complicate quantification, we subsequently employed cryo-electron microscopy (cryo-EM) for a more in-depth analysis.

Cryo-EM averages (Fig. 3a, b) were obtained from many single particles of empty rings, 32-NSP1 and 32-NSP1-S. The averages clearly indicate protein density inside the ring for both 32-NSP1 and 32-NSP1-S, where the NSP1 intensity is higher than for NSP1-S. To quantitatively compare the densities, we calculated circularly averaged intensity profiles and normalized the background value to 0 (Supplementary Figure 9). These profiles indicate that the average density inside the 32-NSP1 is ~2.4-fold higher than for 32-NSP1-S. Rotationally aligned 2D class averages for the empty rings showed 19 circularly distributed density spikes that mutually connect the three radial DNA layers (Supplementary Figure 10), which can be matched to DNA crossovers forming connections between neighboring helices in the DNA ring.

**Molecular dynamics modeling provides Nup density maps**. To obtain microscopic insight into the spatial distribution of the FG-Nups inside the pore, we used a coarse-grained (CG) MD model to simulate the FG-Nups[27,28] (Methods section) and calculated the time-averaged protein density distribution for 32-NSP1 and 32-NSP1-S inside the rings. The average mass density in the 32-NSP1 pore is clearly higher than for 32-NSP1-S (Fig. 3c, d and Supplementary Figure 9c, d). Interestingly, we observe that the NSP1 pores feature a strong spatial variation in protein density (middle panel of Fig. 3d) with a $z$-averaged value of ~50 mg ml$^{-1}$ at the central axis (Supplementary Figure 9c, d). In contrast, the NSP1-S pores show a more uniform protein distribution (bottom panel of Fig. 3d) with a considerably reduced density of ~32 mg ml$^{-1}$ at the central axis (Supplementary Figure 9c, d). We attribute the higher densities of NSP1 to its high percentage of hydrophobic residues relative to charged residues, consistent with expectations[27]. For both NSP1 and NSP1-S, we observe that

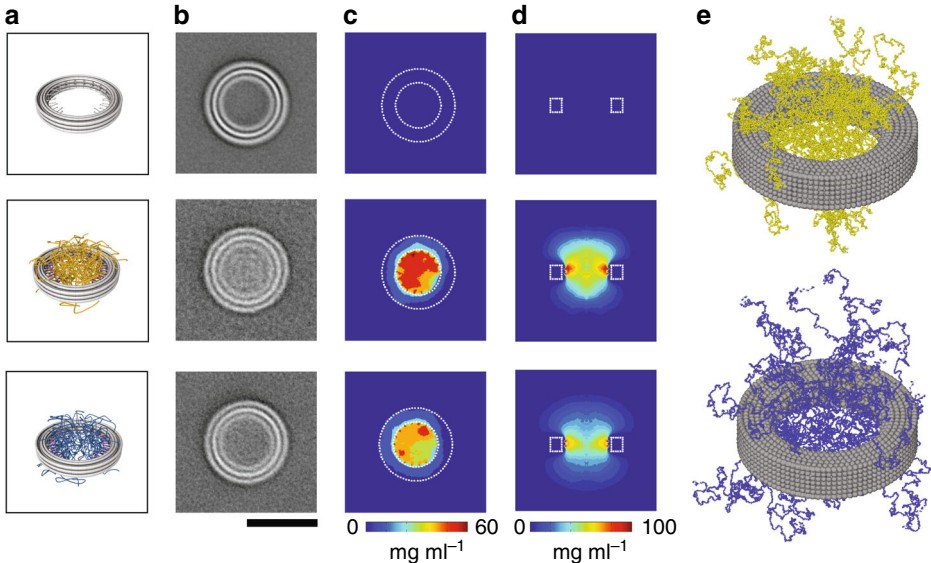

**Fig. 3** Spatial distribution of FG-Nup densities in DNA rings from cryo-EM and MD simulations. **a** Schematic representation of (top to bottom): empty ring, 32-NSP1, and 32-NSP1-S. **b** Corresponding average obtained from aligning and summing individual cryo-EM particles. Number of averaged particles: Ring = 1663, NSP1 = 637, NSP1-S = 1051. See Supplementary Figure 8 for exemplary particles. Scale bar = 50 nm. **c**, **d** Time-averaged mass densities of proteins inside the DNA ring obtained from coarse-grained MD simulations averaged in the z-direction, shown in top view (**c**) and side view (**d**). **e** Exemplary snapshot of MD simulations of 32-NSP1 (top) and 32-NSP1-S (bottom), showing that NSP1-S proteins extend further out than NSP1

proteins are spilling out of the ring (Fig. 3d, e), which likely accounts for the cloudy density on top of the rings seen in the cryo-EM data, which also blurs the circular stripes in the origami TEM images. While the density differences between 32-NSP1 and 32-NSP1-S in the simulations match the trend of the experimental intensities obtained by cryo-EM, the absolute mass ratios quantitatively deviate (2.4 in the measured densities versus 1.2 in the simulated densities integrated over the inner volume of the ring). The difference may be due to excess NSP1 proteins in solution that bind to the rings via hydrophobic interactions with the attached NSP1 proteins and that stay attached upon purification from excess proteins.

**Ion conductance of NPC mimic origami rings on nanopores.** We employed nanopore ionic current measurements[19,20,29–31] to determine the ion conductance of various ring–protein assemblies. DNA origami rings were docked onto solid-state nanopores using the electrophoretic force provided by a 100 mV applied voltage (Fig. 4a, Methods section)[29,30]. Current versus voltage (IV) measurements in 250 mM KCl yielded linear curves (Fig. 4b), indicating the stability of the ring in the docked position[29]. To measure the influence of docked rings on the ionic conductance we docked various ring–protein combinations repeatedly on the same nanopore to avoid pore-to-pore variations (exemplary current traces in Fig. 4c). For rings without proteins we found a reduction in the conductance of 15 ± 5% for both 32 and 8 attachment anchors. (Supplementary Figure 11). A small conductance decrease is expected since the rings partially block the current path in the access region of the solid-state nanopore[29]. To probe for variations in the docking position of a ring on a nanopore, we repeatedly reversed the applied voltage for 10 ms to release and re-dock a single ring multiple times on the same nanopore, which yielded variations of the reduced conductance of ±2% (Supplementary Figure 12). We performed experiments for all ring–protein combinations on four different nanopores in which we docked each variant multiple times to obtain reliable statistics, always including a ring variant without attached

proteins for comparison[31] (Fig. 4d–g, Supplementary Figures 11 and 13).

We found that increasing the number of attached proteins systematically increases the conductance blockade. For instance, 8-NSP1 results in a reduced median conductance of 23 ± 5%, while 32-NSP1 yields a blockage of 34 ± 6% on the same nanopore. Moreover, when varying the type of protein, we found that NSP1 blocks the ionic current more strongly than NSP1-S (20 ± 4% for 8-NSP1 vs. 18 ± 3% for 8-NSP1-S and 35 ± 5% for 32-NSP1 vs. 31 ± 10% for 32-NSP1-S). We can understand these reduced ionic conductances of Nup-filled rings from a simple model that we recently developed (see methods and Supplementary Figures 14–15). The model assumes a critical protein density, above which no ion conductance is supported. From the spatial protein density distribution found in the simulations (Fig. 4h), the ion conductance can then be computed without any further fitting parameters (Methods section), yielding conductance values that compare well with the experimental results (Fig. 4i). While it is gratifying that this simple model captures all trends well, the absolute calculated values are consistently lower than the experimental values which is explained by the fact that experimentally, ions are observed to leak through the DNA ring while such an ionic permeability of the origami structures was not considered in the model.

## Discussion
Taken together, our experiments demonstrate the successful development of a NPC mimic based on a DNA origami ring as a scaffold to position NSP1 and NSP1-S Nups. This approach circumvents major limitations of previous studies by allowing precise control over the number and location of FG-Nup attachment sites which opens the way to high-resolution cryo-EM imaging and transport studies. We find that NSP1 forms a much denser mass distribution compared to NSP1-S, consistent with the reduced hydrophobic interactions in the mutant[23,32].

Our approach opens the way to many more sophisticated future experiments on well-controlled NPC mimics. We anticipate that additional NPC components may be integrated into the

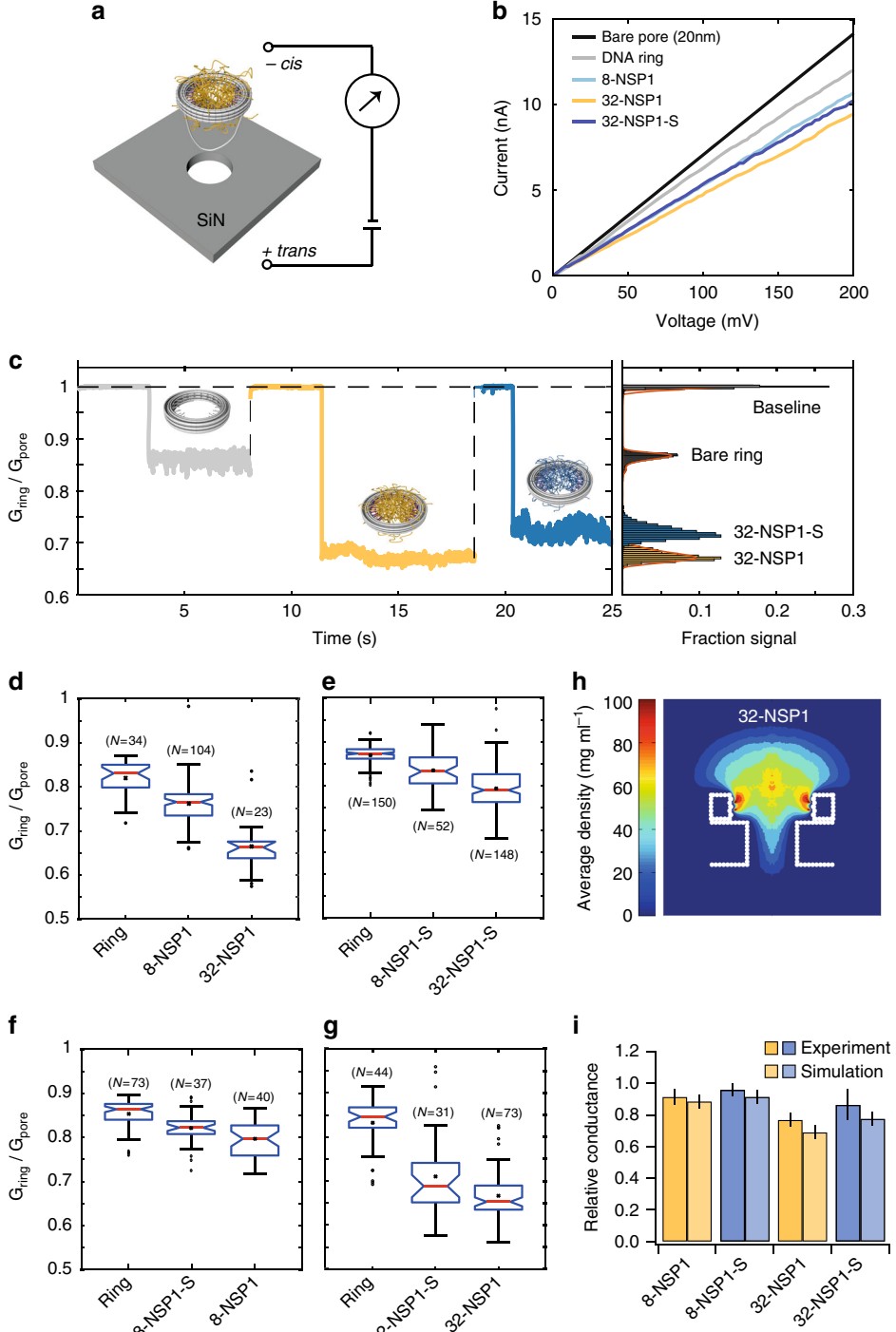

**Fig. 4** Ionic conductance of rings with FG-Nups docked on a solid-state nanopore. **a** Schematic representation of DNA ring that is docking onto a solid-state nanopore. **b** Exemplary current (nA) versus voltage (mV) traces for rings without proteins, 8-NSP1, 32-NSP1 and 32-NSP1-S and the bare nanopore. **c** Exemplary relative conductance ($G_{ring}/G_{pore}$) vs time (s) traces showing the change in conductance upon docking of the ring without protein (gray), 32-NSP1 (yellow), and 32-NSP1-S (blue). **d** Box plot representation of the relative conductance ($G_{ring}/G_{pore}$) for the empty ring, 8-NSP1 and 32-NSP1. **e** Same as (**d**), but for empty ring, 8-NSP1-S and 32-NSP1-S. **f** Same as (**d**), but for empty ring, 8-NSP1 and 8-NSP1-S. **g** Same as (**d**), but for empty ring, 32-NSP1 and 32-NSP1-S. Each of the panels **d–g** represents a different nanopore experiment where a series of rings are probed on one particular solid-state nanopore. In the box plot representation in **d–g**, the blue boxes denote the 25th and 75th percentiles and the red lines represent the median values with the associated wedges representing a 95% confidence interval for the medians (see methods and Supplementary Table 4). **h** Side view (*rz* plane) average density distribution for 32-NSP1 placed on a 20 nm-wide nanopore in a 20 nm thin SiN membrane (see Supplementary Figure 14 for an exemplary simulation snapshot and Fig. S15 for density distributions of other variants). **i** Comparison of experimental reduced conductance values and simulation results (Supplementary Note 2 and Supplementary Tables 5 and 6)

 

DNA origami scaffold, for example, different Nups on different anchoring sites, spatial variation along the axial $z$-direction, as well as the modular stacking of multiple rings with different Nups. Such studies will be of interest for disentangling the mechanism of these fascinating natural gatekeepers to the cell nucleus, as well as potentially for applications involving selective membrane pores, for example in synthetic cell systems. On a more general outlook, we note that intrinsically disordered proteins are notoriously difficult to study, yet increasing evidence is amounting their ubiquitous importance in biology. Our DNA origami-based approach is well suited to help elucidate the role of such proteins in other natural biomolecular assemblies.

## Methods

**Design of DNA origami ring**. The ring was designed in an iterative procedure of using caDNAno v0.2[33] and CanDo[34,35].

**Molecular self-assembly of DNA origami ring**. All reaction mixtures contained single-stranded scaffold DNA at a concentration of 50 nM and oligonucleotide strands (Eurofins MWG, Ebersberg, Germany) at 200 nM each. The reaction buffer includes 5 mM TRIS, 1 mM EDTA, 5 mM NaCl (pH 8) and 20 mM $MgCl_2$. All reaction mixtures were subjected to a thermal annealing ramp using TETRAD (MJ Research, now Biorad) thermal cycling devices. During the annealing, the reaction mixture was exposed to 65 °C for 15 min, then the temperature was decreased with 1° per 2 h down to 40 °C.

**Gel electrophoresis of self-assembly reactions**. Folded DNA nanostructures were electrophoresed on 2% agarose gels containing 0.5× TBE and 11 mM $MgCl_2$ for 2–3 h at 70 V bias voltage in a gel box immersed in an iced water-bath. The electrophoresed agarose gels were stained with ethidium bromide and scanned using a Typhoon FLA 9500 laser scanner (GE Healthcare) at a resolution of 50 µm px$^{-1}$.

**Total internal reflection microscopy**. We annealed the Cy5-oligonucleotides (IDT, Coralville, USA) on origami rings by incubating at a 1:10 ratio of binding spots to oligonucleotides in 250 mM KCl, 10 mM Tris, 1 mM EDTA, and 10 mM $MgCl_2$ at 35 °C for 60 min. We did not purify the origami rings from excess Cy5-oligos as they would be removed during buffer exchange. Flow cells were assembled by sandwiching double-sided tape between PEG passivated microscope quartz slides and cover slips. A small fraction (1:100 ratio) of PEG molecules contained a biotin moiety to facilitate the immobilization and imaging of biotin-labeled DNA origami rings (Supplementary Figure 4). The flow cell was first incubated with 0.1 mg ml$^{-1}$ streptavidin in buffer A (50 mM Tris-HCl pH 8.0, 50 mM NaCl, 10 mM $MgCl_2$) for 1 min. Excess streptavidin was removed with 100 µl of buffer A. Then the Cy5-oligo annealed origami rings of around 50 pM were introduced into the flow cell and incubated for one minute before removing the excess with 100 µl of buffer A. We then introduced an imaging buffer consisting of 40 mM Tris-HCl, pH 8.0, 50 mM NaCl, 10 mM $MgCl_2$, 2 mM trolox. We also included an oxygen scavenging system (0.3 mg ml$^{-1}$ glucose oxidase and 40 µg ml$^{-1}$ catalase with 5% (w/v) glucose as a substrate) for obtaining stable fluorescence from the dye molecules. We used a TIRF-microscope setup that was described earlier[36] for obtaining the bleaching curves of Cy5-labeled rings. We recorded the fluorescence of Cy5 molecules using a 640 nm laser with 30 mW power at a frame rate of 2 Hz until the bright spots reached to the background level. We then plotted the fluorescence intensity from each spot over time. Bleaching of individual fluorophores resulted in a clear step-wise decrease in fluorescence which facilitated us to count the total number of fluorophores in each ring.

**Conjugation of proteins**. Nups proteins were a kind gift of S. Frey and D. Görlich. We used a maleimide-cysteine coupling reaction to conjugate the proteins with an oligonucleotide at the single cysteine at the N-terminal tails of both protein variants. The maleimide-modified oligo was produced by Biomers, Ulm, Germany. The proteins were treated with TCEP prior to incubation with the modified oligonucleotides which was subsequently removed using cut off filters (Merck Millipore). The proteins were incubated with the oligonucleotides in the presence of 5 M GuHCl-PBS overnight. The protein-oligo mixtures were purified from non-attached oligos by size exclusion fast protein liquid chromatography (AKTA) with Superdex 200 increase columns using the same buffer. First, control samples were analyzed using AKTA. One sample contained only the oligo with a maleimide modification ('M-oligo') and one sample contained only NSP1 proteins, to identify the elution peak for each sample. It is important to mention that the use of GuHCl in the running buffer was essential for this step. GuHCl ensures that NSP1 proteins remain in their unfolded conformation and do not interact with each other, forming aggregates which would result in clogging of the SEC column. The protein was stored at −80 °C until it was incubated with the rings.

**Preparation of samples for EM imaging**. After the folding reaction, excess oligonucleotides were subsequently removed by agarose gel extraction followed by a PEG precipitation to increase the concentration[37]. DNA rings were incubated with proteins at a ratio of 1:8 per binding site at a $MgCl_2$ concentration of 20 mM and 2 M GuHCl at room temperature for 6–10 h. Excess proteins were subsequently removed by two rounds of PEG precipitations[37]. The pellet was resuspended in a buffer containing 5 mM Tris, 1 mM EDTA, 5 mM NaCl and 5 mM $MgCl_2$. Shortly before applying the sample to the EM grids the $MgCl_2$ concentration was increased to 20 mM.

**TEM imaging**. Purified sample of DNA rings with or without attached proteins were adsorbed on glow-discharged formvar-supported carbon-coated Cu400 TEM grids (Science Services, Munich, Germany) and stained with a 2% aqueous uranyl formate solution containing 25 mM sodium hydroxide. Imaging was performed using a Philips CM100 electron microscope operated at 100 kV. Images were acquired using a AMT 4 Megapixel CCD camera at a magnification of ×28,500. Tomography tilt series were acquired using a FEI Tecnai Spirit electron microscope operating at 120 kV. Tilt series were acquired using a TemCam-F416 (Tietz, Gauting, Germany) camera at a magnification of ×42,000 with tilt angles between −50° and 50° in steps of 1°. Tomography calculations were performed using IMOD[38].

**Cryo-EM imaging and image processing**. Samples of DNA rings with or without attached proteins (in 20 mM $MgCl_2$, 5 mM tris base, 1 mM EDTA, and 5 mM NaCl) were incubated for 120 s on glow-discharged lacey carbon grids with ultrathin carbon film (Ted Pella, 01824) and vitrified using a freeze-plunging device (Vitrobot Mark IV, FEI). Samples were imaged at liquid nitrogen temperatures using a Titan Krios TEM (FEI) operating at 300 kV with a Falcon II detector (FEI) set to a magnification of ×29,000 and a defocus around −2 µm. 2D averaging was performed with a custom script written in MathWorks MATLAB (R2013b; 8.2.0.701). The particles used for all three averages shown in Fig. 3b were selected by choosing particles with high intensity inside the ring relative to the mean intensity of the particle image. Rationally aligned reference-free class averages (Supplementary Figure 10) were calculated using Relion2[39] and Ctffind v4.0[40] for ctf correction.

**Preparation of samples for nanopore measurements**. All samples were measured at a concentration of 200–300 pM of rings in a buffer containing 250 mM KCl, 50 mM $MgCl_2$ and 10 mM TE. The (8 or 32) rings were incubated in 250 mM KCl, 2.5 M GuHCL, 50 mM $MgCl_2$ and 10 mM TE overnight with oligo-NSP1(-S) in 30× excess per binding site in a shaker at 300 r.p.m. and 35 °C. Next, free oligo-NSP1(-S) was removed by incubating the sample with magnetic beads (MB, 6.25 mg pmol$^{-1}$) for at least 30 min in the shaker at 300 r.p.m. and 35 °C. The final concentration of GuHCl in samples containing rings with NSP1(-S) was 150 mM. At least 150 mM GuHCl was also added to rings without NSP1(-S) to adjust the baseline current correspondingly. The magnetic beads have oligos attached that are complementary to the oligo attached to NSP1(-S).

**Ionic conductance measurements with solid-state nanopores**. Samples of DNA rings at ~200 pM in measurement buffer (250 mM KCl, 10 mM TRIS, 1 mM EDTA and 50 mM $MgCl_2$) were added to the *Cis*-chamber of the flow cell (Fig. 4a). When applying a voltage (100 mV), the electrophoretic force acts on the negatively charged DNA rings and pulls them onto the nanopore. Multiple DNA ring samples were loaded per nanopore experiment for comparable results. The custom made PMMA-flow cell chamber with sample was washed with 3-fold excess buffer between loading samples[19,20]. The current vs. voltage (IV) characteristics of the bare pore were determined, to confirm a linear IV dependence without intercept and the stability of the nanopore. IV-curves were recorded from −200 to +200 mV with steps 2.5 mV (Fig. 4b). Data acquisition was performed at room temperature, with +100 mV applied voltage (unless stated otherwise). Ionic currents were detected using a patch clamp amplifier (Axopatch 200B, Axon Instruments) at 100 kHz bandwidth, digitized with a DAQ card at 500 kHz and recorded with Clampex 9.2 (Axon Instruments).

Efforts to detect translocation events of importer proteins were hampered by a significant level of noise in the ionic current signal upon docking the rings to the nanopores (Supplementary Figure 14). Such translocation measurements may be feasible in future work when using a lipid bilayer instead of a solid-state nanopore.

**Conductance blockade analysis**. Current traces were analyzed with a custom Matlab script. The files were loaded into Matlab and filtered (1 kHz low-pass Gaussian) with Transalyzer[41]. The filtered traces were separated between zaps into ring traces for each DNA ring with zap residues removed. Each ring trace was further analyzed as follows. A histogram was fit to the current trace over time (1000 bins nS$^{-1}$). The histogram was smoothed and peaks were selected (minimal peak distance 0.75 nS peak$^{-1}$) using build-in Matlab functions. The baseline conductance was selected and the ring conductance was calculated from the averaged remaining peaks. Finally, the rings were selected with a baseline within 0.75 nS of the estimated average baseline and a minimal event length (0.3 s). The box plots (Fig. 4d–g) were created with build-in Matlab functions from the fractions of the

ring and baseline conductance for each ring. The box plot notches provide a 95% confidence interval for the median, and confidence interval that are disjoint are different at the 5% significance level. The vertical and horizontal black bars denote the whiskers extending to the most extreme data points; the individual black dots represent outliers. The box plot whiskers in Fig. 4d–g (black) correspond to ~ ±2.7σ[42]. All the medians and respective confidence intervals are tabulated in Supplementary Table 4.

**Salt concentrations**. We note that the interactions between the Nups potentially may depend slightly on the salt concentration and the corresponding screening length. For the cryo-EM measurements, we used 20 mM $MgCl_2$ + 5 mM NaCl which yields a screening length of ~1.5 nm. For the conductivity measurements, we used 250 mM KCl + 50 mM $MgCl_2$, yielding a shorter screening length of ~0.5 nm. The screening length in cells is ~0.8 nm (150 mM monovalent salt concentration).

**Molecular dynamics simulations**. The one bead per amino acid MD model used here accounts for the exact amino acid sequence of the FG-Nups, with each bead centered at the $C_\alpha$ positions of the polypeptide chain[27,28]. The average mass of the beads is 120 Da. Each bond is represented by a stiff harmonic spring potential with a bond length of 0.38 nm[28]. The bending and torsion potentials for this model were extracted from the Ramachandran data of the coiled regions of protein structures. Solvent polarity is incorporated through a distance-dependent dielectric constant, and ionic screening is accounted for through Debye screening with a screening length that is consistent with the salt concentrations used in the cryo-EM and conductance experiments as described in the previous section. The hydrophobic interactions among the amino acids are incorporated through a modified Lennard-Jones potential accounting for hydrophobicity scales of all 20 amino acids through normalized experimental partition energy data renormalized in a range of 0 to 1. For details of the method and its parametrization, the reader is referred to reference[27]. The DNA ring was modeled as a cylinder of height 13.85 nm and diameter of 36 nm constructed from inert beads of diameter 2.6 nm. The DNA ring is modeled in detail and the FG domains are anchored to the scaffold at the specified attachment sites given by the origami design (Fig. 3e, Supplementary Figure 1b-c and Supplementary Figure 14).

MD simulations were carried out using GROMACS 4.5.1. First, the systems were energy minimized to remove any overlap of the amino acid beads. Then the long-range forces were gradually switched on. The simulations were carried out for over $5 \times 10^7$ steps (with the first $5 \times 10^6$ steps ignored for extracting the end-result data), which was found to be long enough to have converged results in the density distribution inside the pores. The time-averaged density calculations presented in the main text were carried out by centering the nanopore in a $100 \times 100 \times 140$ nm box, which was divided into discrete cells of volume $(0.5$ nm$)^3$ and the number density in each cell was recorded as a function of simulation time. Finally, the number density was averaged over the simulation time and multiplied with the mass of each bead to get the time-averaged 3D density profile. The 3D density in cylindrical coordinates $\rho(r,\theta,z)$ was averaged in the circumferential ($\theta$) direction to obtain two-dimensional (2D) $\rho(r,z)$ density plots (as shown in Figs. 3d and 4h). The 3D density was also averaged in the $z$-direction (extending out to $|z| = 25$ nm, where the density was found to be zero) to obtain 2D $\rho(r,\theta)$ density distributions (Fig. 3c). Finally, the radial density distribution $\rho(r)$ was obtained by averaging the 2D $\rho(r,z)$ density maps in the vertical direction ($|z| < 25$ nm), and shown in Supplementary Figure 9c-d. For comparison with the cryo-EM data, we integrated the circularly averaged 2D $\rho(r,z)$ density profile (Fig. 3d) over radii corresponding to the inside of the ring ($r = 18$ nm) and over $|z| < 25$ nm, giving the total mass M of proteins inside the ring.

**Density-based conductance calculation**. We previously developed a model to calculate the conductance from the density of the FG-Nups in a separate study in which NSP1 and NSP1-S were directly attached to solid-state nanopores[43]. Here we briefly recapitulate the model's essentials, before describing its extension to account for the DNA ring. The ionic conductance $G(d)$ for cylindrical bare solid-state (SiN) nanopores of diameter $d$ can be expressed as[44,45]

$$G(d) = \sigma_{bare}\left[4l/\left(\pi d^2\right) + 1/d\right]^{-1} \quad (1)$$

where the first and second terms in the denominator account for the pore resistance and the access resistance, respectively. Here $l = 20$ nm is the height of the pore and $\sigma_{bare}$ is the ionic conductivity through the bare pore. In order to probe the conductance of the nanopores coated with FG-Nups, we developed a density-based conductance relation by assuming that the presence of protein reduces the conductivity in the pore and access region by means of volume exclusion[43]:

$$G(d) = \left[\left(4l/\left(\pi d^2 \sigma_{pore}\right)\right) + \left(1/\left(d\sigma_{access}\right)\right)\right]^{-1} \quad (2)$$

To calculate the effective conductivity $\sigma_{pore}$ for a specific pore diameter $d$, we make use of the radial density distributions $\rho(r)$ of the Nups inside the pore, i.e., averaged over the range $-10$ nm $< z < 10$ nm. The ion conductivity is taken equal to $\sigma_{bare}$ for regions where the Nup density is zero. The conductivity is assumed to decrease linearly with the local protein density as $\sigma(r) = \sigma_{bare}(1 - \rho(r)/\rho_{crit})$, where $\rho_{crit}$ is

taken equal to 85 mg ml$^{-1}$[43], and set to zero at and beyond that critical density. Then by radially integrating $\sigma(r)$, the conductivity of the pore can be calculated as

$$\sigma_{pore} = \left(4/\pi d^2\right)\int_{r=0}^{r=\frac{d}{2}} 2\pi r\sigma(r)\mathrm{d}r. \quad (3)$$

A similar expression is also used to calculate the access conductivity ($\sigma_{access}$), but with the radial density distribution $\rho(r)$ obtained by integrating over $z$-values in the access region, i.e., 10 nm $< |z| < 40$ nm[46]. The conductance results for the bare pore as well as for SiN nanopores coated with NSP1 or NSP1-S were shown to be in excellent agreement with the experimentally observed sconductances[43].

In the study presented here, we extended the model to FG-Nups tethered inside a DNA ring which is placed on top of a bare nanopore (Supplementary Figure 14). The conductance experiments for the Nup-coated DNA ring on top of a bare nanopore were carried out at a salt concentration of 250 mM KCl + 50 mM $MgCl_2$ leading to a bulk conductivity of $\sigma_{bare} = 4.30 \pm 0.13$ nS nm$^{-1}$ (from bulk conductivity measurements at these conditions). For the system with the DNA ring shown in Supplementary Figure 14, z = 0 nm is chosen to be the center of the DNA ring so that the SiN nanopore region corresponds to $-27$ nm $< z < 7$ nm (Supplementary Figure 15). In Supplementary Figure 15 it can be observed that outside the SiN nanopore the protein has non-zero density toward the ring side (top) and zero density on the other (bottom) side. Therefore, the access resistance contains contributions from the top side with non-zero protein density in the region $-7$ nm $< z < 33$ nm and from the bottom side in the region $z < -27$ nm with zero density. Therefore, we modified the access resistance term in Eqn. 2 to differentiate between the top and bottom access resistance, resulting in the conductance relation for FG-Nup-coated DNA rings placed on a SiN pore, as

$$G(d) = \left[4l/\left(\pi d^2 \sigma_{pore}\right) + 1/(2d\sigma_{access}) + 1/(2d\sigma_{bare})\right]^{-1} \quad (4)$$

The results of these calculations are shown in Fig. 4i and Supplementary Table 5.

**Data availability**. The data that support the findings of this study are available from the corresponding authors upon reasonable request.

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

## Acknowledgements

We gratefully acknowledge Steffen Frey and Dirk Görlich for providing the NPC proteins. We thank Klaus Wagenbauer for support with the FEI Titan microscope, Pascal Hauenstein for support in the wetlab, and Florian Praetorius for scaffold DNA preparations. We thank Sabrina Wistorf for assistance with protein-oligo conjugation. This work was supported by NanoNextNL (program 07 A.05 to C.D.), the ERC Advanced Grant SynDiv (grant number 669598 to C.D.); the Netherlands Organization for Scientific Research (NWO/OCW) (part of the Frontiers of Nanoscience program, to C.D.), the Zernike Institute for Advanced Materials (University of Groningen) and the UMCG to A.M., the ERC Starting Grant to H.D. (grant 256270 to H.D.), the Deutsche Forschungsgemeinschaft through grants provided within the Gottfried-Wilhelm-Leibniz Program to H.D., the Excellence Clusters CIPSM (Center for Integrated Protein Science Munich to H.D.), NIM (Nanosystems Initiative Munich to H.D.), Technische Universität München (TUM) Institute for Advanced Study to H.D., and the Graduate School IGSSE to H.D.

## Author contributions

P.K., A.N.A., D.L.T., A.M., and E.B. performed the research. C.D. and H.D. designed the research. P.K. designed the DNA ring. P.K. and E.B. prepared the DNA rings, collected and analyzed the EM data. A.N.A. and D.L.T. collected and analyzed the nanopore data. A.N.A. and M.G. collected and analyzed the TIRF data. A.N.A. and J.vd.T. carried out oligo-protein conjugation and AKTA purification. A.M. performed the MD simulations under the supervision of P.O. P.K., A.N.A., A.M., H.D., and C.D. wrote the manuscript. All authors commented on the manuscript.
