## [Peer Review File · Nature Communications]

Reviewers' comments:

Reviewer #1 (Remarks to the Author):

The manuscript by Ketterer, Ananth, et al describes construction of a DNA origami scaffolded for studying intrinsically disordered proteins of the nuclear pore complex, as the title indicates. The approach is clever and versatile, and the work is very well carried out. I have only very technical suggestions. Unfortunately though, the final accomplishment is also technical. I don't find any specific scientific conclusion. The setup is obviously a step on the way to study selective transport mimicking the nuclear pore, but the conductance measurements only indicate that the origami ring has docked to the nanopore. There is no indication of gating or conformational response relating to macromolecular transport. Under the circumstances I think that publication of the ring construction itself is better suited to a more nanotechnology-oriented journal such as Small or ACS Nano.

Minor remarks:

1. In the 3rd paragraph "based on DNA origami based scaffolds". based appears twice.
2. On page 4, I would have liked to see the length of the double-helical separator domain specified in the text.
3. In figure 2, it is hard to see the difference between empty rings and the 32-NSP1-S specimens. This might be a printing issue, but it impacts the bar graph in panel h. The number of pores examined should be specified. Does supplementary figure 6 show a representative group or the whole dataset? How many pores were evaluated by tomography?
4. Similarly, how many pores entered the statistics for 2D averaging from cryo-EM data?
5. Is there some independent verification that docking occurs according to the scheme of supplementary figure 12a?

Reviewer #2 (Remarks to the Author):

FG nucleoporins (Nups) are intrinsically disordered proteins that line the central channel of the nuclear pore complex. They mediate nucleocytoplasmic exchange, probably by phase separation. FG-Nups are challenging to study in *in vitro*, because the native structure of the NPC provides a grafting scaffold that enforces a very unique local concentration and orientation *in situ* but thus far cannot be reconstituted *in vitro*. Ketterer, Ananth et al. present a DNA origami scaffold that mimics nuclear pore complex shape. They graft the FG-Nup Nsp1 onto this scaffold to study its biophysical properties under close to native conditions. The approach is laudable and the first of its kind that allows precise control of FG-Nup copy number and spatial distribution.

Overall, the authors present a convincing and very interesting first proof of concept. They show that NSP1 fills up the space within the ring with dense material. A previously characterized mutant of NSP1 that is less cohesive has a reduced tendency to do so and appears to occupy a wider space. This is in line with EM image quantification, MD simulations and conductivity measurements.

Although the novel functional insights of this study are somewhat limited, the potential of this method is very exciting and an important contribution to several scientific fields. This paper is suitable for the broad audience of Nat Comm and I recommend it for publication.

Some comments:

Why was the decoration efficiency measured with Cy5-oligos and not with Cy5-Nsp1-oligo conjugates? The latter would be more convincing because whether or not locally enriched Nsp1 interferes with hybridization efficiency of additional molecules is difficult to estimate. Vice versa, it could also attract additional Nsp1 molecules by phase separation thus creating an 'over-decoration'. Could simple DNA and protein concentration assays clarify this aspect?

Although probably credible, the 'low level' image quantification presented in Figure 2f-h is not really state of the art. Even simple area intensity measurements as implemented in e.g. for light microscopy in ImageJ would provide a more objective measure.

Terminology: a 'cryo electron micrograph' is an individual image that usually contains many 'single particles' that can be aligned and combined into an 'average'. Terms like 'Cryo-EM average micrographs', 'single-particle cryo-EM micrographs' or 'average micrographs' are rather unusual.

The class average shown in figure S10 is a bit arbitrary without being able to see other representative classes.

34 nm are only somewhat 'similar to the inner diameter of the central channel', that actually is a little wider, but for a first proof of concept this is fine.

Reviewer #3 (Remarks to the Author):

The paper reports the manufacturing of a nanopore decorated with the DNA origami ring. The origami scaffold enables controlled attachment of desired number of intrinsically disordered proteins of the Nuclear Pore Complex (FG nups) to serve in the future as an NPC mimic and a versatile tool to study intrinsically disordered proteins in general.

The technical achievements are impressive and the system is comprehensively characterized. However, I have several questions/comments, mostly pertaining to the quantification of the differences between origami nanopores decorated with NSP-1 and the mutant NSP-1S.

1. Figure 2. I feel that the "observer-based" estimates of the protein density within nanopores (essentially eye-balling) is inadequate and a more rigorous quantification method should be employed. For instance, Fig 3 and Fig S9 report the actual measurements of the protein density within the pore. Why panels f-h in Fig 2 are necessary?

2. Figure 3b. Is the averaging performed over ALL nanopores, or only those deemed to be "medium" or "high" density?

3. Figure 4 d-i. I assume that the blue shape denote the standard error in the estimation of the mean, while black bars denote the actual variance of the measurements (this should be explained)? Similarly to the above - are averages performed on all observed rings or only on those deemed to contain protein? It is somewhat dangerous to compare averages over highly divergent measurements. It looks from the scatter of the values of the conductivity that different rings vary greatly in the amount of protein. In this respect, presented, the difference between 32-NSP-1 and 32-NSP1-S looks within the margin of error. I think the limitations of these comparisons should be clearly discussed. For instance - do the values of conductivity of individual pores correlate with the amount of protein in them; can this be reproduced in simulations?

4. Figure S9a-c. What is the origin of these oscillations? More puzzlingly, why do averaged density profiles also exhibit these oscillations? Are they due to high density ordering near the scaffold?

Reviewer #1

The manuscript by Ketterer, Ananth, et al describes construction of a DNA origami scaffolded for studying intrinsically disordered proteins of the nuclear pore complex, as the title indicates. The approach is clever and versatile, and the work is very well carried out. I have only very technical suggestions. Unfortunately though, the final accomplishment is also technical. I don't find any specific scientific conclusion. The setup is obviously a step on the way to study selective transport mimicking the nuclear pore, but the conductance measurements only indicate that the origami ring has docked to the nanopore. There is no indication of gating or conformational response relating to macromolecular transport. Under the circumstances I think that publication of the ring construction itself is better suited to a more nanotechnology-oriented journal such as Small or ACS Nano.

We thank the referee for acknowledging the quality and versatility of our work. We note that we in fact find a number of interesting results on NSP1 and its mutants in our study, namely how these key proteins spatially distribute within the rings, and how this does compare to molecular dynamics simulations. These findings are new and will be of interest to the community.

However, we do acknowledge that the major focus of our work is on the entirely new approach. We believe this DNA origami-based route to construct biomimetic nuclear pores – which in itself involved significant technical challenges in the design and characterization that we report here - present a radical new way for studying this important protein complex, which is bound to attract major attention and will be of interest to the broad readership of Nature Communications, as is also mentioned by Reviewer 2 and Reviewer 3.

Minor remarks:

1. In the 3rd paragraph "based on DNA origami based scaffolds". based appears twice.

The manuscript was edited accordingly.

2. On page 4, I would have liked to see the length of the double-helical separator domain specified in the text.

This information was added as requested.

3. In figure 2, it is hard to see the difference between empty rings and the 32-NSP1-S specimens. This might be a printing issue, but it impacts the bar graph in panel h. The number of pores examined should be specified. Does supplementary figure 6 show a representative group or the whole dataset? How many pores were evaluated by tomography?

We decided to take out panel f-h in Figure 2 because of the subjective evaluation method. Please also see our response to R3 who raised a similar point.

For completeness: The number of pores evaluated in the analysis were Ring = 909, NSP1 = 1012, NSP1-S = 1039. In total, we obtained 3 tomograms containing 2 pores each for the sample with NSP1.

4. Similarly, how many pores entered the statistics for 2D averaging from cryo-EM data?

We thank the referee for pointing out that this information was missing. The number of particles has now been added to the caption of figure 3 (Ring = 1663, NSP1 = 637, NSP1-S = 1051).

5. Is there some independent verification that docking occurs according to the scheme of supplementary figure 12a?

The scheme depicted in Fig. S12 is based on experience from past experiments (Plesa et al. ACS Nano 8, 35, 2014) where an origami structure was docked onto a solid-state nanopore, as guided by a long DNA tail at its bottom. Concretely, the redocking experiments were carried out exactly similar to the recapture experiments previously published on DNA origami nanoplates (See supporting information section 4, in Plesa et al., ACS Nano 8, 35, 2014). Furthermore, the docking scheme is supported by our conductivity data summarized in Fig. 4, that show that increasing the number of proteins reduces the ionic conductance as expected.

Reviewer #2

FG nucleoporins (Nups) are intrinsically disordered proteins that line the central channel of the nuclear pore complex. They mediate nucleocytoplasmic exchange, probably by phase separation. FG-Nups are challenging to study in vitro, because the native structure of the NPC provides a grafting scaffold that enforces a very unique local concentration and orientation in situ but thus far cannot be reconstituted in vitro. Ketterer, Ananth et al. present a DNA origami scaffold that mimics nuclear pore complex shape. They graft the FG-Nup Nsp1 onto this scaffold to study its biophysical properties under close to native conditions. The approach is laudable and the first of its kind that allows precise control of FG-Nup copy number and spatial distribution.

Overall, the authors present a convincing and very interesting first proof of concept. They show that NSP1 fills up the space within the ring with dense material. A previously characterized mutant of NSP1 that is less cohesive has a reduced tendency to do so and appears to occupy a wider space. This is in line with EM image quantification, MD simulations and conductivity measurements.

Although the novel functional insights of this study are somewhat limited, the potential of this method is very exciting and an important contribution to several scientific fields. This paper is suitable for the broad audience of Nat Comm and I recommend it for publication.

We thank the referee for the enthusiastic evaluation and emphasizing the interest and importance of our work.

Some comments:

Why was the decoration efficiency measured with Cy5-oligos and not with Cy5-Nsp1-oligo conjugates? The latter would be more convincing because whether or not locally enriched Nsp1 interferes with hybridization efficiency of additional molecules is difficult to estimate. Vice versa, it could also attract additional Nsp1 molecules by phase separation thus creating an 'over-decoration'. Could simple DNA and protein concentration assays clarify this aspect?

We chose to measure the decoration efficiency with Cy5-oligos because of technical problems with Cy5-Nsp1-oligo conjugates (limitations in synthesis of the oligo with Cy5 and maleimide side groups by multiple suppliers; inefficient conjugation of Cy5-oligo to Nsp1; low yield in purification).

We took care to prevent over-decoration by the use of 2.5M GuHCL in the attachment of the Nsp1-oligo conjugates to origami rings. This information was inadvertently missing in the methods section, and now has been added under "Preparation of samples for nanopore measurements".

Although probably credible, the 'low level' image quantification presented in Figure 2f-h is not really state of the art. Even simple area intensity measurements as implemented in e.g. for light microscopy in ImageJ would provide a more objective measure.

We decided to take out panel f-h in Figure 2; please see our response to R3 who raised a similar point.

Terminology: a 'cryo electron micrograph' is an individual image that usually contains many 'single particles' that can be aligned and combined into an 'average'. Terms like 'Cryo-EM average micrographs', 'single-particle cryo-EM micrographs' or 'average micrographs' are rather unusual.

The manuscript was edited accordingly.

The class average shown in figure S10 is a bit arbitrary without being able to see other representative classes.

We include other reference-free class averages with a significant number of particles of the same measurement below. The number of particles in each class is indicated below the images. The first class average below is the same as in Fig. S10. We obtained similar class averages in multiple measurements. We added the additional class averages to Fig. S10 and the number of particles in each class to the caption.

34 nm are only somewhat 'similar to the inner diameter of the central channel', that actually is a little wider, but for a first proof of concept this is fine.

We replaced 'similar' with 'approximating' in the manuscript.

Reviewer #3

The paper reports the manufacturing of a nanopore decorated with the DNA origami ring. The origami scaffold enables controlled attachment of desired number of intrinsically disordered proteins of the Nuclear Pore Complex (FG nups) to serve in the future as an NPC mimic and a versatile tool to study intrinsically disordered proteins in general.

The technical achievements are impressive and the system is comprehensively characterized. However, I have several questions/comments, mostly pertaining to the quantification of the differences between origami nanopores decorated with NSP-1 and the mutant NSP-1S.

We thank the referee for the kind words and highlighting the versatility of our approach.

1. Figure 2. I feel that the "observer-based" estimates of the protein density within nanopores (essentially eye-balling) is inadequate and a more rigorous quantification method should be employed. For instance, Fig 3 and Fig S9 report the actual measurements of the protein density within the pore. Why panels f-h in Fig 2 are necessary?

Panel f-h were meant as a qualitative illustration. We admit that the classification is subjective and hence we now decided to take panels f-h in Figure 2 out in the revised manuscript. We adjusted Fig. S6 accordingly to show only exemplary particles without classification.

2. Figure 3b. Is the averaging performed over ALL nanopores, or only those deemed to be "medium" or "high" density?

We thank the referee for pointing out this missing information. Particles of all three samples were selected by the high intensity inside the ring relative to the overall mean intensity of the particle image. Note that we performed this selection also for the sample with no protein in order to treat all samples equally. We added a short description on how the particles were selected in the methods section in the supplementary information.

3. Figure 4 d-i. I assume that the blue shape denote the standard error in the estimation of the mean, while black bars denote the actual variance of the measurements (this should be explained)?

We apologize for the missing information on the box plot formats. Here is the requested information: The red bar indicates the median; the wedges of the blue box at the red bar are the 95% confidence interval for the median; the extremes of each blue box represent the 25th and 75th percentile; the vertical and horizontal black bars denotes the whiskers extending to the most extreme data points; the individual

black dots denote the outliers. We have extended our explanation on the box plot details both in the caption of Figure 4 and in the supplementary information under “Conductance blockade analysis”.

Note that we have corrected Fig 4d-g in the revised manuscript to rectify a small formatting error where the whiskers of box plots were previously not connected to the upper and lower extremes. Also, for clarity we removed the black crosses (mean) from the box plots, as we do not discuss these in the manuscript.

Similarly to the above - are averages performed on all observed rings or only on those deemed to contain protein?

The nanopore conductance analysis is performed on all observed rings for a given sample. The rings were not selected for conductance, as this would yield a biased result. Events were only selected for a sufficient docking duration ($> 0.3\text{sec}$) on the nanopore as mentioned in the supplementary information.

It is somewhat dangerous to compare averages over highly divergent measurements. It looks from the scatter of the values of the conductivity that different rings vary greatly in the amount of protein. In this respect, presented, the difference between 32-NSP-1 and 32-NSP1-S looks within the margin of error. I think the limitations of these comparisons should be clearly discussed.

We note that the 95% confidence intervals for the median (i.e. the wedges of the blue boxes in Fig 4d-g) are disjoint for all compared samples (e.g. 32-NSP1 and 32-NSP1-S), such that all medians are significantly different at the 5% significance level. For clarity, we have now included a new supplementary table 1 in the supplementary information containing the median and 95% confidence interval for all box plot data from Fig 4d-g.

For instance - do the values of conductivity of individual pores correlate with the amount of protein in them; can this be reproduced in simulations?

Yes, the conductance of individual pores correlates with the amount of proteins in the DNA rings as shown in Fig 4. Furthermore, the molecular dynamics simulation data reproduced the experimental data well, as shown in Fig 4i and discussed in the SI section “Density-based conductance calculation”.

4. Figure S9a-c. What is the origin of these oscillations? More puzzlingly, why do averaged density profiles also exhibit these oscillations? Are they due to high density ordering near the scaffold?

The oscillations between radii 17 and 29nm in Figure S9a-b arise from individual layers of DNA in the ring. This is well known for origami structures. For comparison, we show an excerpt from a previous publication below in which similar oscillations were observed for DNA origami structures in the same honeycomb lattice configuration.

Part of figure 2 in Douglas et al., Nature 459.7245 (2009): 414-418.

In Fig. S9c, the protein densities that were obtained from our simulations (as shown in Fig. S9d) were added to a density profile of the ring that was calculated from a .pdb file derived from a CanDo simulation (see caption of Fig. S9).

Reviewers' Comments:

Reviewer #2:

Remarks to the Author:

The authors have addressed my comments. I recommend this manuscript for publication.

Reviewer #3:

Remarks to the Author:

The authors have addressed my concerns and I have no further comments.